# The Meta-Position of Phe^4^ in Leu-Enkephalin Regulates Potency, Selectivity, Functional Activity, and Signaling Bias at the Delta and Mu Opioid Receptors

**DOI:** 10.3390/molecules24244542

**Published:** 2019-12-12

**Authors:** Robert J. Cassell, Krishna K. Sharma, Hongyu Su, Benjamin R. Cummins, Haoyue Cui, Kendall L. Mores, Arryn T. Blaine, Ryan A. Altman, Richard M. van Rijn

**Affiliations:** 1Department of Medicinal Chemistry and Molecular Pharmacology, College of Pharmacy, Purdue University, West Lafayette, IN 47907, USA; rcassell@purdue.edu (R.J.C.); Su147@purdue.edu (H.S.); kmores@purdue.edu (K.L.M.); harri374@purdue.edu (A.T.B.); 2Department of Medicinal Chemistry, The University of Kansas, Lawrence, KS 66045, USA; sharma.979@osu.edu; 3Department of Chemistry, Purdue University, West Lafayette, IN 47907, USA; bcummins96@gmail.com; 4College of Wuya, Shenyang Pharmaceutical University, Shenyang 110016, China; Cuihy1@shanghaitech.edu.cn; 5Purdue Institute for Drug Discovery, Purdue University, West Lafayette, IN 47907, USA; 6Purdue Institute for Integrative Neuroscience, Purdue University, West Lafayette, IN 47907, USA

**Keywords:** Leu-enkephalin, beta-arrestin, mu opioid receptor, delta opioid receptor, biased signaling, DADLE, ischemia, plasma stability

## Abstract

As tool compounds to study cardiac ischemia, the endogenous δ-opioid receptors (δOR) agonist Leu^5^-enkephalin and the more metabolically stable synthetic peptide (d-Ala^2^, d-Leu^5^)-enkephalin are frequently employed. However, both peptides have similar pharmacological profiles that restrict detailed investigation of the cellular mechanism of the δOR’s protective role during ischemic events. Thus, a need remains for δOR peptides with improved selectivity and unique signaling properties for investigating the specific roles for δOR signaling in cardiac ischemia. To this end, we explored substitution at the Phe^4^ position of Leu^5^-enkephalin for its ability to modulate receptor function and selectivity. Peptides were assessed for their affinity to bind to δORs and µ-opioid receptors (µORs) and potency to inhibit cAMP signaling and to recruit β-arrestin 2. Additionally, peptide stability was measured in rat plasma. Substitution of the meta-position of Phe^4^ of Leu^5^-enkephalin provided high-affinity ligands with varying levels of selectivity and bias at both the δOR and µOR and improved peptide stability, while substitution with picoline derivatives produced lower-affinity ligands with G protein biases at both receptors. Overall, these favorable substitutions at the meta-position of Phe^4^ may be combined with other modifications to Leu^5^-enkephalin to deliver improved agonists with finely tuned potency, selectivity, bias and drug-like properties.

## 1. Introduction

Leu^5^-enkephalin (Tyr^1^-Gly^2^-Gly^3^-Phe^4^-Leu^5^, Figure 1) is an endogenous opioid peptide produced in vertebrate species including rodents, primates and humans [1,2,3,4] that results from the metabolism of proenkephalin or dynorphin [5]. Pharmacologically, Leu^5^-enkephalin agonizes the δ opioid receptor (δOR) with moderate selectivity over the µ opioid receptor (µOR), but does not significantly interact with the κ opioid receptor [6]. As a neurotransmitter in pain circuits, Leu^5^-enkephalin possesses antinociceptive properties [7], whereas peripherally, the peptide demonstrates cardiovascular effects [8]. Over the last two decades, improved pharmacological characterization of opioid pathways has revealed that activation of an opioid receptor can trigger two distinct pathways, β-arrestin-dependent or β-arrestin-independent (i.e., G protein-mediated) and that these pathways differentially modulate antinociception and side effect profiles [9]. Despite the increasing number of studies that implicate δOR mediated β-arrestin recruitment with various (patho)physiological effects, such as tolerance [10], alcohol intake [11,12] and δOR agonist-induced seizures [10], the role of β-arrestin recruitment towards δOR-induced cardioprotection remains unclear.

From a translational perspective, peptide-based probes provide an ideal tool for studying the cardioprotective effects of the δOR, given their low brain penetration. While numerous enkephalin-like peptides have been synthesized that interact with excellent potency and selectivity for δORs and µORs [13], a majority of studies [14] investigating δOR involvement in ischemia have utilized the synthetic peptide d-Ala^2^, d-Leu^5^-enkephalin (**DADLE**, Figure 1) [15,16], because **DADLE** possesses improved proteolytic stability and improved selectivity for δORs over µORs relative to Leu^5^-enkephalin [6,17]. However, **DADLE**’s discovery in 1977 [18], predated identification of β-arrestin as a modulator of opioid signaling [14,19,20,21], With the use of contemporary cellular assays it is now apparent that **DADLE** pharmacologically signals similarly to Leu^5^-enkephalin, though it recruits β-arrestin 2 (arrestin 3) slightly more efficaciously. Given the similarities between **DADLE** and Leu^5^-enkephalin, it is unclear to what degree β-arrestin recruitment contributes to or detracts from these peptides’ in vivo cardioprotective efficacy, and new analogs with distinct pharmacoloogical profiles are necessary to probe these contributions.

To better investigate the role of δOR-mediated β-arrestin signaling in ischemic protection, the development of δOR selective agonists that have either low, intermediate or high β-arrestin is desired; however, reports of δOR selective peptide-based biased ligands remain limited. Recently, the naturally occurring peptides rubiscolin-5 and -6 (Figure 1) were classified as G protein-biased (β-arrestin 2 efficacy = 15% and 20%, respectively; δOR bias factor = 2.0) δOR selective peptides, albeit with only micromolar potencies [22]. While a number of synthetic peptides display nanomolar potencies at δORs, such as aza-β-homoleucine-enkephalin (β-arrestin 2 efficacy = 64%; δOR bias factor = 5.2) [23] and Dmt-Tic analogs [24], these peptides all recruit β-arrestin 2 far more efficaciously than the rubiscolins (UFP-512: cAMP potency 0.4 nM, β-arrestin 2 potency = 20 nM, efficacy = 60%) [23,24,25]. Additionally, these latter biased compounds, including aza-β-homoleucine-enkephalin and Dmt-Tic-Gly-NH-Ph, have at best 10-fold selectivity for δORs over µORs and actually super-recruit β-arrestin 2 at µOR [23,24], which is likely to cause adverse, undesired in vivo effects [26,27]. As such, our goal was to identify novel and potent δOR peptide agonists with varying degrees of β-arrestin recruitment efficacy, and that importantly have improved δOR selectivity over µOR while limiting β-arrestin recruitment at µOR.

A collection of previously published structure activity relationship studies on Leu^5^-enkephalin directed us to Phe^4^ as a position that can modulate δOR and µOR potency and selectivity (Figure 2A) [28]. Specifically, ortho and para substitutions on Phe^4^ can modulate binding affinity and selectivity between opioid receptors [28], and halogenation of the para-position of Leu^5^-enkephalin and endomorphin 1 in particular appears to enhance δOR affinity while reducing µOR affinity [28]. Corroborating research has further shown that halogenation of the para position of Phe^4^ of [D-Pen^2^,D-Pen^5^]enkephalin (DPDPE) enhanced δOR selectivity, potency, peptide stability, central nervous system (CNS) distribution, and antinociceptive potency compared to unsubstituted DPDPE [29,30,31]. Thus, halogenation at Phe^4^ may provide Leu^5^-enkephalin derivatives that can be used to study not only cardiac but also cerebral ischemia [32,33]. We herein report structure-activity-relationship trends at the meta-position of Phe^4^ of Leu^5^-enkephalin (Figure 2B), demonstrating that substitution at this position variously regulates δOR and µOR affinity and G-protein activity, enables the fine-tuning of β-arrestin 2 recruitment to both δOR and µOR, and further increases the plasma stability of the derived peptides. Combined, these features provide a clear direction for designing the next-generation of Leu^5^-enkephalin analogs with well-defined biases and improved stabilities.

## 2. Results and Discussion

### 2.1. Design Considerations

To probe the meta position of Phe^4^, we initially considered known structure-activity-relationship trends at the ortho and para positions of this residue. Considering that halogenated substituents at these positions perturbed binding affinity, δOR selectivity, and stability properties of the Leu^5^-enkephalin [28], we initially hypothesized that meta-halogenated analogs might similarly perturb the parent scaffold (Figure 3A, **1a**–**1d**). An additional set of analogs bearing electron-donating (**1e**–**1f**) and -withdrawing groups (**1g**–**1i**) would further probe interactions at this site, including the electronic character of the Phe^4^ ring. Finally, pyridine analogs (**1j**–**1l**) would present H-bond accepting contacts about the ring, as well as provide analogs that present dipoles at similar vectors as to previously successful halogenated substituents.

### 2.2. Solid Phase Synthesis of Peptides

All peptides were synthesized using a rapid solid phase peptide synthesis protocol on an automated peptide synthesizer using an Fmoc protection strategy [34,35] and *N*,*N*′-diisopropylcarbodiimide and oxyma as the coupling reagents (Figure 3B). Fmoc-Leu-Wang resin was utilized as a starting template for this synthetic protocol. All coupling steps and Fmoc-deprotection steps were carried out at 70 °C under an atmosphere of N_2_. Cleavage from the resin was performed using TFA/triisopropylsilane/H_2_O. Purification of the synthesized peptides was performed by reverse-phase high performance liquid chromatography (RP-HPLC), and analysis of purity was performed using ultra-performance liquid chromatography (UPLC). All desired peptides were obtained in ≥95% purity before submitting for pharmacological evaluation.

### 2.3. Pharmacological Characterization

To characterize our substituted analogs, we assessed binding affinity by competition radioligand binding, G protein potency and efficacy using a cAMP GloSensor assay and β-arrestin 2 recruitment via PathHunter assays at both the δOR and µOR. Using Leu^5^-enkephalin as well as DAMGO (for µOR), as reference compounds, substitution of the meta position of Phe^4^ (**1a**–**1i**) generally increased binding affinity for the δOR (Table 1, *K*_i_ = 0.023–0.93 nM; Leu^5^-enkephalin = 1.26 nM) and µOR (*K*_i_ = 0.059–0.98 nM; Leu^5^-enkephalin = 1.7 nM). The improved binding affinity correlated with improved functional activation of the G protein pathway at both the δOR and µOR, while providing near-full agonist activity at the δOR (Appendix A, 92–100% efficacy) relative to Leu^5^-enkephalin, and mostly near-full agonist activities at the µOR (Appendix A, 85–105% efficacy; Leu^5^-enkephalin = 100% efficacy vs. DAMGO).

At the δOR, meta-substituted analogs recruited β-arrestin 2 with a range of potencies (Table 2, EC_50_: 0.56–49 nM; Leu^5^-enkephalin = 8.9 nM) with near-full efficacies (Table 2, 91–130%; Leu-enkephalin = 100%), and likewise at the µOR, analogs recruited β-arrestin 2 with a broad range of potencies (Table 2, EC_50_: 36–589 nM; Leu-enkephalin = 977 nM) and efficacies (Table 2, 60–96% relative to DAMGO; Leu-enkephalin = 60%; Figure 4). Interestingly, **DADLE** recruited β-arrestin 2 more efficaciously than Leu^5^-enkephalin at both δOR (126%) and µOR (99%). Picoline analogs **1j**–**1l**, generally showed reduced affinity at both receptors (Table 1, δOR *K*_i_ = 6.2–33 nM; µOR *K*_i_ = 9–158 nM), which further correlated with decreased G protein activation of the receptors (Table 2, δOR IC_50_ = 4.6–48 nM; µOR IC_50_ = 41–302 nM). At the δOR, the low affinity of pyridyl-substituted analogs correlated with low β-arrestin 2 recruitment (Table 2, EC_50_ = 100–1122 nM; 84–98% efficacy), though these substitutions drastically affected β-arrestin 2 recruitment though the µOR (Table 2, EC_50_ = 1.3–41.6 µM; 36–70% efficacy).

Within these overall trends, specific analogs show unique profiles. Meta− chlorination, −bromination, or −iodination (**1b**–**1d**) produced high-affinity analogs of Leu^5^-enkephalin that super-recruited β-arrestin 2 and that drastically increased functional selectivity (Table 3). These increases in affinity at δOR linearly correlate (R^2^ = 0.998) with the atomic van der Waals radii (H, 1.20 Å; Cl, 1.77 Å; Br, 1.92 Å; I, 2.06 Å) [36], and might suggest a halogen bonding interaction similar to those previously explored at the ortho position of Phe^4^ [28]. However, the distinct vectors at which a sigma-hole would present from the ortho vs. meta positions suggest that the specific residues engaged through meta-substitution are likely distinct from those previously identified through ortho-substitution [28], and future computational collaborative work might enable a better understanding of the unique binding interactions between the δOR and meta-halogenated analogs of Leu^5^-enkephalin. Notably, the meta-chlorinated and −brominated analogs (**1b**, and **1c**) are 500–900-fold more potent in cAMP assays at δOR than at µOR, despite exhibiting little differences in binding affinity at δOR relative to µOR (Table 1). Interestingly, the meta-Cl analog (**1b**) had stronger bias towards G protein-signaling at δOR (Table 3, bias factor 1.6), but towards β-arrestin 2 recruitment at µOR relative to Leu^5^-enkephalin (Table 3, bias factor 0.004) as well as DAMGO (Table 3, bias factor 0.3), which provides a unique pharmacological profile for future uses. In contrast, meta-F and −CN and −NO_2_-substitutions (**1a,h,i**) improved δOR functional selectivity (Table 3), but lost β-arrestin 2 potency at δOR relative to Leu^5^-enkephalin. Additionally, meta-OMe and −NO_2_ substitution (**1f**, **1i**) provided both potent and biased analogs, with G protein coupling activities comparable to Leu^5^-enkephalin (Table 2, IC_50_ = 0.14–0.47 nM vs. 1.02 nM), but with improved bias factors relative to Leu^5^-enkephalin at both δOR (Table 3, bias factor 6.9–7.5) and µOR (Table 3, bias factor 8.4–11.1). Of these two analogs, the -NO_2_-substituted analog **1i** exhibited higher δOR functional G protein selectivity (Table 3, 104-fold δOR selectivity) relative to the −OMe analog **1f** (Table 3, 58-fold δOR selectivity). Though pyridyl-substituted analogs (**1j**–**1l**) showed poor potency and efficacy for the δOR and µOR relative to Leu^5^-enkephalin (Table 2), the 3- and 4-pyridyl analogs (**1k**–**1l**) showed strong bias at µOR (Table 3, bias factor 17–331), when compared to the full agonist DAMGO. However, if instead of DAMGO, Leu^5^-enkephalin was used as the reference compound, most analogs, with the exception of **1k** and **1l** lost G protein bias (Table 3, bias factor < 1), because the analogs generally were more potent and efficacious than Leu^5^-enkephalin in recruiting β-arrestin 2 at µOR (Table 2), with the exception of **1k** (Table 2, 36% recruitment efficacy). Given that exogenous Leu^5^-enkephalin analogs in vivo would compete with endogenous Leu^5^-enkephalin and not DAMGO, our results highlight the limitations of interpreting bias factors (particularly using an unnatural compound, such as DAMGO, as a reference) and the associated risk of using bias factor as a major driver of lead optimization.

Compared to **DADLE**, the Leu^5^-enkephalin analogs **1b** and **1c** displayed a similar signaling bias profile, but were more potent and selective for δOR, and thus may provide a better tool compound to target δORs for the treatment of cardiac ischemia. However, compounds **1b** and **1c** are also more potent than **DADLE** at recruiting β-arrestin 2 to µOR, resulting in a lower bias factor (Figure 4). In contrast, analog **1i** has a favorable pharmacology relative to **DADLE**, with similar selectivity as **1c**, but improved G-protein bias at δOR and µOR relative to **DADLE**. Given the potential adverse effects associated with strong µOR-mediated β-arrestin 2 recruitment, analog **1k** is potentially useful, as it exhibited weak β-arrestin 2 recruitment (36% efficacy) and concomitant strong bias factor for G-protein signaling. However, cAMP potency for **1k** at δOR was more than one log unit weaker than **DADLE**, and thus it may be necessary further optimize the ligand by increasing δOR potency in the cAMP assay, while retaining the low µOR β-arrestin 2 recruitment efficacy (Figure 4).

### 2.4. Stability

The stability of all compounds to Sprague Dawley rat plasma was assessed to study the influence of the meta− and picoline/pyridine-substitutions at Phe^4^ relative to Leu^5^-enkephalin (Table 4). For this parent compound, the predominant known routes of metabolism and clearance occur through cleavage of Tyr^1^-Gly^2^ by aminopeptidase *N* [37,38], and of Gly^3^-Phe^4^ by angiotensin converting enzyme [39], and combined, the plasma metabolism occurs with a half-life (*t*_1/2_) of < 10 min. In general, meta substituted Phe^4^ analogs exhibited improved plasma stability compared with Leu^5^-enkephalin with half-lives typically >20 min. The 3-fluoro derivative (**1a**) was the most stable analog with a half-life of 82.3 min. From UPLC-mass spectrometry analysis of degradation fragments, meta-substitution did not greatly impede the proteolysis at the Tyr^1^-Gly^2^ site, but instead slowed digestion at the Gly^3^-Phe^4^ site (Table 4). Thus, the improved stability of our Phe^4^-substituted analogs presumably derived from perturbation/deceleration of angiotensin-converting enzyme activity. Picoline peptides also displayed improved stability though interestingly, UPLC-mass spectrometry analysis indicated that degradation of all pyridyl-substituted analogs (**1j**–**1l**) predominantly occurred through Tyr^1^-Gly^2^ as opposed to meta-substituted analogs **1a**–**1i** that degraded through cleavage of Gly^3^-Phe^4^.

## 3. Methods

### 3.1. Synthetic Chemistry-General Considerations

Unless specified, all chemicals were purchased from commercial sources and used without further purification. All solvents used for synthesis were of analytical grade and used without further purification. Proton nuclear magnetic resonance (^1^H-NMR) spectra, and carbon nuclear magnetic resonance (^13^C-NMR) spectra were recorded on a Bruker AVIII 500 AVANCE spectrometer with a CPDUL cryoprobe (500 and 126 MHz, respectively) or a Bruker DRX 500 spectrometer (500 and 126 MHz, respectively). Fluorine nuclear magnetic resonance (^19^F-NMR) spectra were recorded on a Bruker AVIII 500 AVANCE spectrometer with a CPDUL BBFO cryoprobe (376 MHz) or a Bruker DRX 500 spectrometer (376 MHz). Chemical shifts (δ) for protons are reported in parts per million (ppm) downfield from tetramethylsilane, and are referenced to proton resonance of solvent residual peak in the NMR solvent (MeOD-*d*_4_: δ = 4.87 ppm or DMSO-*d*_6_: δ = 2.50 ppm). Chemical shifts (δ) for carbon are reported in ppm downfield from tetramethylsilane, and are referenced to the carbon resonances of the solvent residual peak (MeOD-*d*_4_: δ = 49.1 ppm or DMSO-*d*_6_: δ = 39.52 ppm). ^19^F-NMR chemical shifts (δ) are reported in ppm with respect to added internal standard: PhCF_3_ (δ = −63.72 ppm). NMR data are represented as follows: chemical shift (ppm), multiplicity (s = singlet, d = doublet, dd = doublet of doublets, t = triplet, q = quartet, sept = septet, m = multiplet), coupling constant in Hertz (Hz), and integration. Exact mass determinations were obtained by using electrospray ionization using a time of flight mass analyzer (Waters LCT Premiere).

Peptides were synthesized using an Aapptec Focus XC automated peptide synthesizer coupled with a heating system using a solid phase peptide synthesis protocol using Fmoc chemistry. Preparative RP-HPLC was performed using an appropriate column and solvent system (described below) and final purity of peptides was determined by UV area % from UPLC analysis. Peptides were purified by Teledyne ISCO EZ Prep system on RediSep^®^ C18 Prep column (30 × 250 mm, 100 Å). Purity analysis of final peptides was carried out using a Waters UPLC Aquity system equipped with a PDA e*λ* detector (200 to 400 nm) and a HSS T3 C18 column, (1.8 μM, 2.1 × 50 mm column), using one of two methods. Protocol A: gradient elution of 2% MeCN with 0.1% formic acid in H_2_O to 98% MeCN over 2.5 min, then holding at 98% MeCN for 3 min at a flow rate of 0.6 mL/min. Protocol B: gradient elution of 2% MeCN with 0.1% formic acid in H_2_O to 98% MeCN over 2.5 min, then holding at 98% MeCN for 1 min at a flow rate of 0.7 mL/min at 40 °C. Plasma stability was also assessed using the above UPLC column and solvent gradient program using a Waters H class Plus Acquity UPLC system coupled with a QDa detector using protocol B.

### 3.2. Synthesis of Peptides

Peptides were synthesized using a solid phase peptide synthesis protocol using an Aapptec Focus XC automated peptide synthesizer coupled with a heating system using the Fmoc chemistry and Wang resin as solid support [34]. To prepare the resin for synthesis, a reaction vessel equipped with a sintered glass bottom was charged with Fmoc-Leu-Wang resin (0.2 mM), and swelled in a mixture of dichloromethane and DMF (1:1) for 15 min. The resin was then transferred to a peptide synthesizer reaction vessel. The resin was deprotected twice using 20% piperidine in DMF for 5 min at 70 °C. Subsequently, an Fmoc-protected amino acid was double coupled with the Leucine-wang resin by treating with *N*,*N*′-diisopropylcarbodiimide (3.0 equiv., 0.2 M in DMF) and Oxyma (3.0 equiv., 0.2 M in DMF) at 70 °C for 8 min. Completion of coupling reactions was monitored by a Kaiser’s test for the initial peptide [40]. Each coupling was followed by removal of the Fmoc group using 20% piperidine in DMF at 70 °C for 5 min and repeated once. The cycle of the Fmoc removal and coupling was repeated with subsequent Fmoc-protected amino acids to generate the desired resin-bound peptide. Cleavage of the peptide from resin and concomitant deprotection of the side chain protecting groups was carried out by shaking in TFA/triisopropylsilane/H_2_0 (95/2.5/2.5; 5 mL), at ambient temperature for 3 h. Subsequent filtration afforded the peptide in the filtrate and the volume was reduced to 0.2 mL. Then, the crude peptides were precipitated by adding cold diethyl ether, and the crude peptides were then purified by RP-HPLC. Synthesized peptides were characterized by NMR, and the high-resolution mass spectroscopy.

### 3.3. Materials Used in the Cellular Signaling Assays

Leu^5^-enkephalin and forskolin, were purchased from Sigma-Aldrich (St. Louis, MO USA). (d-Ala^2^, N-MePhe^4^, Gly-ol) enkephalin (DAMGO) was purchased from Tocris Bioscience (Minneapolis, MN, USA). Radiolabels were from Perkin Elmer (Waltham, MA, USA).

### 3.4. Cell Culture and Biased Signaling Assays

cAMP inhibition and β-arrestin 2 recruitment assays were performed as previously described [12]. In brief, for cAMP inhibition assays HEK 293 (Life Technologies, Grand Island, NY, USA) cells were transiently transfected in a 1:3 ratio with FLAG-mouse δOR, or HA-mouse µOR and pGloSensor22F-cAMP plasmids (Promega, Madison, WI, USA) using Xtremegene9 (Sigma). Two days post-transfection cells (20,000 cells/well, 7.5 µL) were seeded in low volume Greiner 384-well plates (#82051-458, VWR, Batavia, IL, USA) and were incubated with Glosensor reagent (Promega, 7.5 µL, 2% final concentration) for 90 min at room temperature. Cells were stimulated with 5 µL drug solution for 20 min at room temperature prior to stimulation with 5 µL forskolin (final concentration 30 µM) for an additional 15 min at room temperature. For β-arrestin recruitment assays, CHO-human µOR PathHunter β-arrestin 2 cells or CHO-human δOR PathHunter β-arrestin 2 cells (DiscoverX, Fremont, CA, USA) were plated (2,500 cells/well, 10 µL) one day prior to stimulation with 2.5 µL drug solution for 90 min at 37 °C/5% CO_2,_ after which cells were incubated with 6 µL cell PathHunter assay buffer (DiscoverX) for 60 min at room temperature as per the manufacturer’s protocol. Luminescence for each of these assays was measured using a FlexStation3 plate reader (Molecular Devices, Sunnyvale, CA, USA).

### 3.5. Radioligand Binding Assay

For the binding assay 50 µL of a dilution series of peptide was added to 50 µL of 3.3 nM [3H]DPDPE (*K*_d_ = 3.87 nM) or 2.35 nM of [3H]DAMGO (*K*_d_ = 1.07 nM) in a clear 96 well plate. Next, 100 µL of membrane suspension containing 7 µg protein was added to the agonist wells and incubated for 90 min at room temperature. The reaction mixture was then filtered over a GF-B filter plate (Perkin Elmer) followed by four quick washes with ice-cold 50 mM Tris HCl. The plate was dried overnight, after which 50 µL scintillation fluid (Ultimagold uLLT) was added and radioactivity was counted on a Packard TopCount NXT scintillation counter. All working solutions were prepared in a radioligand assay buffer containing 50 mM Tris HCl, 10 mM MgCl_2_, and 1 mM ethylenediaminetetraacetic acid at pH 7.4.

### 3.6. Calculation of Bias Factor

Bias factors were calculated using the operational model equation in Prism 8 to calculate Log R (τ/KA) (Appendix A) as previously described [12]. Subsequently, bias factors were calculated using Leu^5^-enkephalin as reference compound for δOR and using either DAMGO or Leu^5^-enkephalin as reference compound for µOR, respectively. Leu^5^-enkephalin and DAMGO were more potent in the cAMP (G protein) assay than in the β-Arrestin 2 recruitment assay, and thus were not unbiased, but rather G protein-biased to begin with. A bias factor > 1 meant that the agonist was more G protein-biased than the reference compound; A bias factor < 1 meant that the agonist was less G protein-biased than the reference compound.

### 3.7. Data and Statistical Analysis

All data are presented as means ± standard error of the mean, and analysis was performed using GraphPad Prism 8 software (GraphPad Software, La Jolla, CA). For in vitro assays, nonlinear regression was conducted to determine pIC_50_ (cAMP) or pEC_50_ (β-arrestin 2 recruitment). Technical replicates were used to ensure the reliability of single values, specifically each data point for binding and β-arrestin recruitment was run in duplicate, and for the cAMP assay in triplicate. The averages of each independent run were counted as a single experiment and combined to provide a composite curve in favor of providing a ‘representative’ curve. In each experimental run, a positive control/reference compound was utilized to allow the data to be normalized and to calculate the log bias value. A minimum of three independent values were obtained for each compound in each of the cellular assays.

### 3.8. Assessment of Plasma Stability

Sprague Dawley rat plasma containing K_2_-ethylenediaminetetraacetic acid (Innovative Research, MI, USA) was transferred into 300 µL aliquots and stored at −20 °C until use. The plasma stabilities of Leu^5^-enkephalin and its synthesized analogs were determined in plasma diluted to 50% with saline (isotonic sodium chloride solution; 0.9% *w*/*v*) [41]. An aliquot of the resulting solution (25 μL) was incubated at 37 °C for 15 min before the addition of a solution of a Leu^5^-enkephalin analog (25 μL of a 100 μM isotonic NaCl solution; 0.9% *w*/*v*). After adding analog, an aliquot of the mixture was removed at each indicated time point (0, 5, 10, 15, 20, 30, 60, 120, 240 min) and immediately quenched with 100 μL of a methanol solution containing 20 μM Fmoc-Leu-OH as an internal standard. The resulting peptide solutions were centrifuged at 13,000 rpm for 15 min at 4 °C on tubes equipped with 2000 MW filtrate tubes (Sartorius, USA). Then, the filtrate was transferred into vials and a 5 μL sample of the resulting solution was analyzed on an UPLC system coupled with a QDa detector. For quantitative determination, area-under-the-curve for the peaks corresponding to the UV chromatogram was measured for both the Leu^5^-enkephalin analog and the internal standard. Determination of half-life (*t*_1/2_) was carried out by using the GraphPad Prism one-phase decay method.

## 4. Conclusions

Meta-substitutions of Phe^4^ of Leu^5^-enkephalin were generally well tolerated and certain substitutions improved affinity, potency, δOR selectivity and stability of this endogenous opioid. The generated pharmacological data herein may aid computational modeling efforts to reveal ligand-receptor interactions at δOR and µOR that will guide the development of novel peptides with tuned selectivity and signaling profiles. The novel Leu^5^-enkephalin analogs have superior δOR selectivity over µOR relative to DADLE, the gold-standard peptide for studying the role of δOR in cardiac ischemia. Additionally, the analogs generally have lower β-arrestin recruitment efficacy at µOR, which could further reduce potential adverse in vivo effects. Finally, relative to **DADLE**, the meta-substituents tune the bias profile at δOR (either more or less biased towards G-protein signaling), and the resulting tool compounds should be useful for investigating the importance of δOR mediated β-arrestin signaling in the peptides cardioprotective effects.

## Figures and Tables

**Figure 1 molecules-24-04542-f001:**
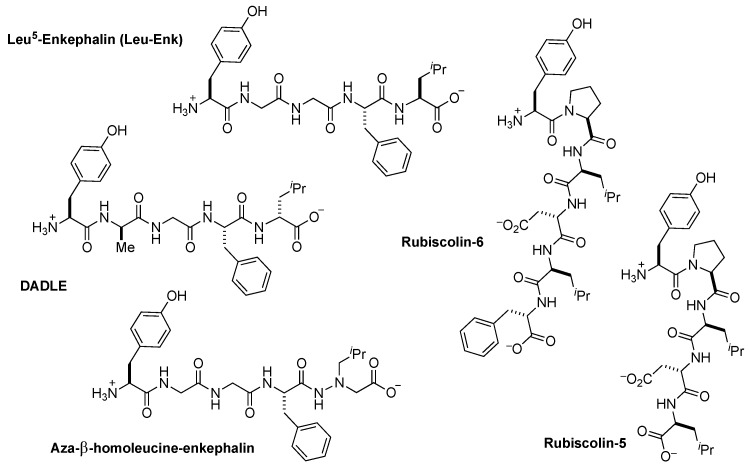
Overview of unbiased (Leu^5^-enkephalin and **DADLE**) and biased (Aza-β-Homoleucine-Enkephalin, Rubiscolin-5 and -6) δOR peptides.

**Figure 2 molecules-24-04542-f002:**
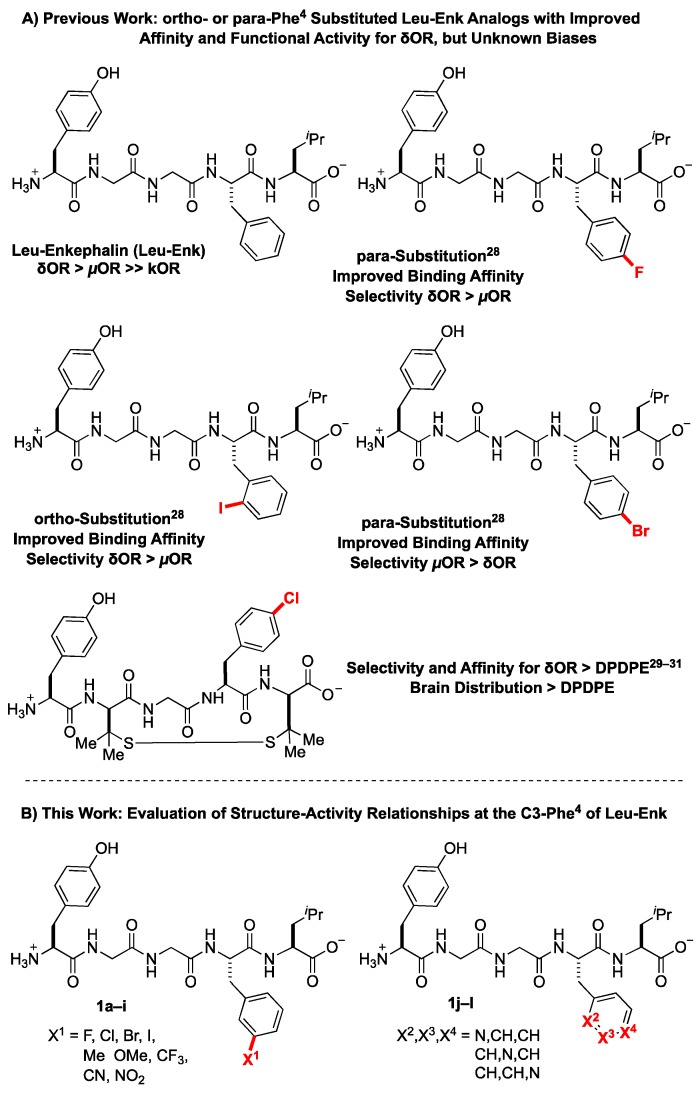
Substituents of Phe^4^ of Leu^5^-enkephalin Affect Pharmacodynamic, Stability, and Distribution Properties.

**Figure 3 molecules-24-04542-f003:**
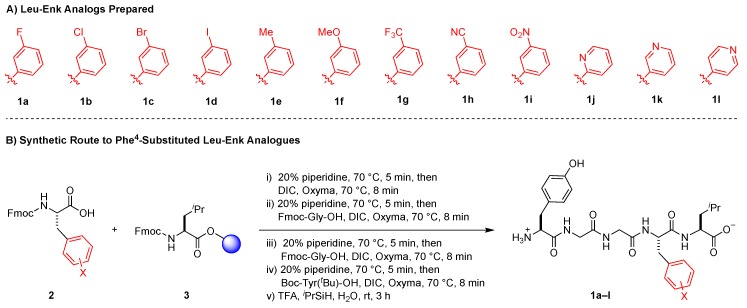
Peptide Synthesis of Phe^4^-Substituted Analogs of Leu^5^-enkephalin.

**Figure 4 molecules-24-04542-f004:**
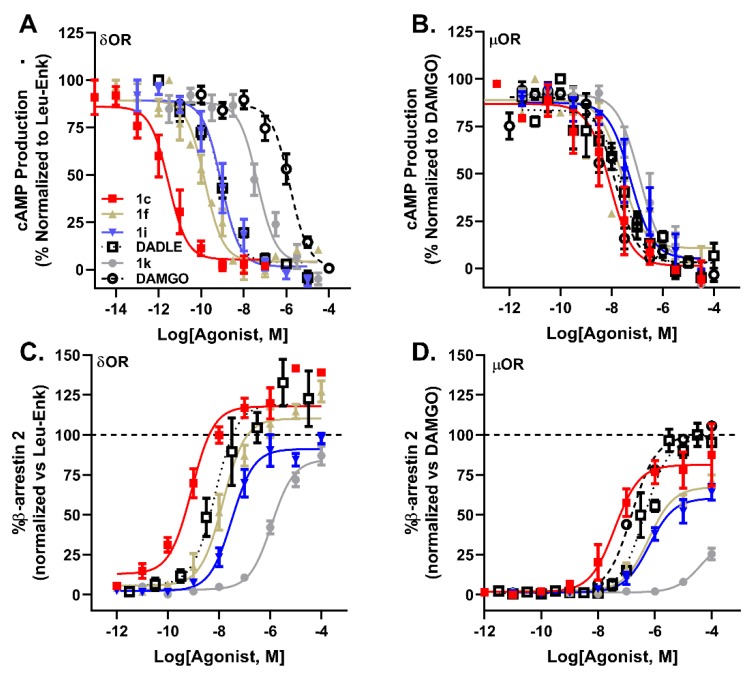
Modifications of Phe^4^ of Leu^5^-enkephalin Produces Analogs with Divergent and Distinct Signaling Profiles. Inhibition of cAMP production by **1c** (■), **1f** (▲), **1i** (▼), **1k** (●), Leu^5^-enkephalin (□) and DAMGO (◯) in HEK cells expressing δOR (**A**) and µOR (**B**). Recruitment of β-arrestin 2 by **1c**, **1f**, **1i**, **1k**, Leu-Enk and DAMGO in CHO cells expressing δOR (**C**) and µOR (**D**).

**Table 1 molecules-24-04542-t001:** Meta Substituted Phe^4^ Analogs of Leu^5^-enkephalin Increase Affinity at δOR and µOR.

	*pK*_i_ ± SEM (δOR)	*K_i_* (nM)	*pK*_i_ ± SEM (µOR)	*K_i_* (nM)	Binding Selectivity (δOR vs µOR)
**1a** (F)	9.48 ± 0.1	0.33	9.12 ± 0.4	0.76	2.3
**1b** (Cl)	9.87 ± 0.1	0.13	10.14 ± 0.2	0.072	0.55
**1c** (Br)	10.35 ± 0.2	0.045	9.90 ± 0.3	0.13	2.9
**1d** (I)	10.64 ± 0.2	0.023	9.86 ± 0.5	0.14	6.1
**1e** (Me)	9.86 ± 0.1	0.14	9.86 ± 0.1	0.14	1.0
**1f** (OMe)	9.31 ± 0.1	0.49	9.07 ± 0.1	0.85	1.7
**1g** (CF_3_)	9.93 ± 0.1	0.12	10.23 ± 0.3	0.059	0.49
**1h** (CN)	9.17 ± 0.3	0.68	9.52 ± 0.1	0.30	0.44
**1i** (NO_2_)	9.03 ± 0.2	0.93	9.01 ± 0.1	0.98	1.05
**1j** (2-pyr)	8.21 ± 0.1	6.17	8.04 ± 0.2	9.12	1.5
**1k** (3-pyr)	7.48 ± 0.1	33.1	6.80 ± 0.1	158	4.8
**1l** (4-pyr)	7.69 ± 0.1	20.4	7.41 ± 0.1	38.9	1.9
**DADLE**	9.01 ± 0.1	0.98	8.80 ± 0.1	1.58	1.6
Leu^5^-enkephalin	8.90 ± 0.1	1.26	8.77 ± 0.1	1.70	1.3
DAMGO	-	-	9.01 ± 0.1	0.98	-

All compounds were tested in three independent trials.

**Table 2 molecules-24-04542-t002:** Meta-substituted Phe^4^ Analogs of Leu^5^-enkephalin Display Enhanced δOR and µOR Potency for cAMP Inhibition and β-arrestin 2 Recruitment, But Vary in β-arrestin Recruitment Efficacy.

	cAMP				β-Arrestin 2					
Compound	pIC_50_ ± SEM (δOR)	IC_50_ (nM)	pIC_50_ ± SEM (µOR)	IC_50_ (nM)	pEC_50_ ± SEM (δOR)	EC_50_ (nM)	δOR Efficacy (% ± SEM)	pEC_50_ ± SEM (µOR)	EC_50_ (nM)	µOR Efficacy (% + SEM)
**1a** (F)	9.47 ± 0.2	0.39	8.10 ± 0.2	7.94	8.01 ± 0.2	9.77	107 ± 9	6.24 ± 0.1	575	75 ± 13
**1b** (Cl)	10.66 ± 0.3	0.022	7.52 ± 0.1	30.2	8.77 ± 0.2	1.70	130 ± 10	7.17 ± 0.2	67.6	96 ± 8
**1c** (Br)	10.52 ± 0.3	0.030	8.00 ± 0.4	10	9.25 ± 0.2	0.56	116 ± 7	7.45 ± 0.2	35.5	79 ± 6
**1d** (I)	10.61 ± 0.2	0.025	8.37 ± 0.3	4.27	8.95 ± 0.2	1.12	111 ± 7	7.33 ± 0.2	46.8	70 ± 3
**1e** (Me)	10.41 ± 0.3	0.039	8.43 ± 0.3	3.72	8.46 ± 0.2	3.46	105 ± 5	6.73 ± 0.2	186	71 ± 3
**1f** (OMe)	9.84 ± 0.3	0.14	8.09 ± 0.3	8.12	7.93 ± 0.2	11.7	106 ± 5	6.23 ± 0.1	589	67 ± 3
**1g** (CF_3_)	9.97 ± 0.4	0.11	8.61 ± 0.4	2.45	8.39 ± 0.2	4.07	108 ± 9	7.25 ± 0.2	56.2	71 ± 3
**1h** (CN)	9.46 ± 0.3	0.35	7.92 ± 0.3	12.0	7.77 ± 0.2	17.0	91 ± 5	6.8 ± 0.2	158	68 ± 4
**1i** (NO_2_)	9.33 ± 0.3	0.47	7.31 ± 0.4	49.0	7.31 ± 0.2	49.0	96 ± 6	6.27 ± 0.1	537	60 ± 6
**1j** (2-pyr)	8.34 ± 0.2	4.57	7.39 ± 0.4	40.7	7.00 ± 0.2	100	92 ± 13	5.87 ± 0.1	1349	70 ± 4
**1k** (3-pyr)	7.32 ± 0.2	47.9	6.63 ± 0.4	234	5.95 ± 0.1	1122	85 ± 6	4.38 ± 0.1	41687	36 ± 7
**1l** (4-pyr)	7.52 ± 0.2	30.2	6.52 ± 0.3	302	6.40 ± 0.2	398	98 ± 6	4.43 ± 0.1	37153	64 ± 11
**DADLE**	9.09 ± 0.2	0.81	7.45 ± 0.1	35.5	7.92 ± 0.4	12.0	126 ± 10	6.33 ± 0.1	468	99 ± 3
Leu^5^-enkephalin	8.99 ± 0.1	1.02	7.34 ± 0.2	45.7	8.05 ± 0.1	8.91	100	6.01 ± 0.1	977	61 ± 3
DAMGO	5.91 ± 0.2	1230	7.82 ± 0.1	15.1	<5		-	6.80 ± 0.1	158	100

**Table 3 molecules-24-04542-t003:** Meta-substituted Phe^4^ Analogs of Leu^5^-enkephalin Display a Range of Selectivity and Bias Profiles.

Compound	G Protein Selectivity (δOR *vs* µOR)	Bias Factor		
		δOR	µOR (DG = Ref)	µOR (LE = Ref)
**1a** (F)	20	1.8	22.2	0.36
**1b** (Cl)	1372	3.2	0.3	0.004
**1c** (Br)	333	1.3	1.2	0.02
**1d** (I)	171	3.5	3.2	0.05
**1e** (Me)	95	4.3	11.5	0.19
**1f** (OMe)	58	7.5	8.4	0.14
**1g** (CF_3_)	22	2.0	4.3	0.07
**1h** (CN)	34	6.8	5.4	0.09
**1i** (NO_2_)	104	6.9	11.1	0.18
**1j** (2-pyr)	8.9	4.0	17.2	0.27
**1k** (3-pyr)	4.9	6.7	331.1	5.89
**1l** (4-pyr)	10	2.2	63.7	1.08
**DADLE**	44	0.5	2.3	0.035
Leu^5^-enkephalin	45	1	4.9	1
DAMGO	0.012	-	1	0.003
Aza-β-homoleucine ^¶^	9.9 [23]	5.2 ^¶^	-	1.2
Rubiscolin-5	-	2.0 [22]	-	-

^¶^ from reference 26 using BRET (β-arrestin 2) and EPAC (cAMP) assays. DG = DAMGO, LE = Leu^5^-enkephalin.

**Table 4 molecules-24-04542-t004:** Rat Plasma Stability of Leu^5^-enkephalin and Its Analogs.

Compound	Half-Life (min)	95% CI	Degradation Products (First Appearance)
Leu^5^-enkephalin	9.4	6.3−4.5	Gly-Gly-Phe-Leu (5 min) Phe-Leu (10 min)
**1a** (F)	82.3	68.0–102.6	Gly-Gly-(meta-F)Phe-Leu (5 min) (meta-F)Phe-Leu (60 min)
**1b** (Cl)	37.8	26.5–56.6	Gly-Gly-(meta-Cl)Phe-Leu (5 min) (meta-Cl)Phe-Leu (30 min)
**1c** (Br)	21.5	12.7–38.7	Gly-Gly-(meta-Br)Phe-Leu (5 min) (meta-Br)Phe-Leu (15 min)
**1d** (I)	13.2	11.4–15.2	Gly-Gly-(meta-I)Phe-Leu (5 min) (meta-I)Phe-Leu (15 min)
**1e** (Me)	39.5	30.0–53.5	Gly-Gly-(meta-Me)Phe-Leu (5 min) (meta-Me)Phe-Leu (20 min)
**1f** (OMe)	46.1	30.4–75.6	Gly-Gly-(meta-OMe)Phe-Leu (5 min) (meta-OMe)Phe-Leu (20 min)
**1g** (CF_3_)	44.5	35.0–59.1	Gly-Gly-(meta-CF_3_)Phe-Leu (5 min) (meta- CF_3_)Phe-Leu (30 min)
**1h** (CN)	33.0	24.0–47.0	Gly-Gly-(meta-CN)Phe-Leu (5 min) (meta-CN)Phe-Leu (20 min)
**1i** (NO_2_)	28.8	16.8–55.8	Gly-Gly-(meta-NO_2_)Phe-Leu (5 min) (meta- NO_2_)Phe-Leu (20 min)
**1j** (2-pyr)	26.8	15.4–53.7	(2-pyridyl)Ala-Leu (5 min) Gly-Gly-(2-pyridyl)Ala-Leu (30 min)
**1k** (3-pyr)	54.0	29.0–127.0	(3-pyridyl)Ala-Leu (10 min) Gly-Gly-(3-pyridyl)Ala-Leu (20 min)
**1l** (4-pyr)	78.1	47.5–165.4	(4-pyridyl)Ala-Leu (10 min) Gly-Gly-(4-pyridyl)Ala-Leu (30 min)

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
