# Peer review of "The Meta-Position of Phe4 in Leu-Enkephalin Regulates Potency, Selectivity, Functional Activity, and Signaling Bias at the Delta and Mu Opioid Receptors"

_molecules, 2019, doi:10.3390/molecules24244542_

Round 1

Reviewer 1 Report

This is a very complete pharmacological study on biased opioid receptor agonists.  

In this paper, the authors are developing new agonists for the opioid
receptor with bias related to the beta arrestin pathway. They perform
a detailed pharmacological investigation of the compounds and identify
bias with respect to G protein signalling and with changes in selectivity
for the different types of opioid receptors.

it is a relevant subject as not many biased agonists have been previously
characterised and it gives more options for future research. The paper us very well written, with attention to detail and all the data are well presented and clear.

The conclusions are consistent and well reasoned. The paper could be improved with some experiments in relevant cell systems but I overall recommend it for publication.

Author Response

We thank the reviewer for evaluating our manuscript and providing critical feedback. 

We appreciate the suggestion of testing the novel peptides in more relevant cell systems. A current hurdle preventing us from testing this for the current manuscript is that the arrestin assay is carried out in a commercial cell system. We are in the process of creating an arrestin assay to run in H9C2 cardiomyocyte cells, which are known to express DOR (but not MOR or KOR), however this is not yet operational.

Reviewer 2 Report

In an effort to develop δOR peptides with improved selectivity and unique
signaling properties for δOR signaling studies in cardiac ischemia, authors explored substitution at the Phe4 position of Leu5-enkephalin for its ability to modulate receptor function and selectivity including ability to recruit β-arrestin 2. Authors present experimental evidence of improved agonists with finely tuned potency, selectivity, bias and drug like properties of these substitutions. The study aim is important and the results are interesting and provide good tools for δOR signaling studies in cardiac ischemia.

In general the results are sound and discussions are straightforward. I have just one comment.

Major comments:

Authors identified substitutions with halogen provide greater selectivity, as result of halogen bonding. In that case, authors should discuss more about the structural basis of these substitutions; how these replacements affect in residue-ligand interactions, by comparing the results with the findings from other similar studies.

Author Response

We thank the reviewer for providing constructive feedback on our manuscript.

We have expanded our discussion to highlight the tight correlation (R=0.999) we observe between the affinity of the halogenated peptides for δOR relative to the Van der Waals radii of the halogens. The new discussion can be found on Page 7, lines 160-167

Round 2

Reviewer 2 Report

Authors revised the manuscript well and I have no further comments.